# Hyperphantasia: A Benchmark for Evaluating the Mental Visualization Capabilities of Multimodal LLMs

**Mohammad Shahab Sepehri**[*]    **Berk Tinaz**[†]    **Zalan Fabian**[‡]    **Mahdi Soltanolkotabi**[§]
Department of Electrical and Computer Engineering
University of Southern California, Los Angeles, CA, USA
[*]`sepehri@usc.edu`   [†]`tinaz@usc.edu`
[‡]`fabian.zalan@gmail.com`   [§]`soltanol@usc.edu`

## Abstract

Mental visualization, the ability to construct and manipulate visual representations internally, is a core component of human cognition and plays a vital role in tasks involving reasoning, prediction, and abstraction. Despite the rapid progress of Multimodal Large Language Models (MLLMs), current benchmarks primarily assess passive visual perception, offering limited insight into the more active capability of internally constructing visual patterns to support problem solving. Yet mental visualization is a critical cognitive skill in humans, supporting abilities such as spatial navigation, predicting physical trajectories, and solving complex visual problems through imaginative simulation. To bridge this gap, we introduce Hyperphantasia, a synthetic benchmark designed to evaluate the mental visualization abilities of MLLMs through four carefully constructed puzzles. Each puzzle is procedurally generated and presented at three difficulty levels, enabling controlled analysis of model performance across increasing complexity. Our comprehensive evaluation of state-of-the-art models reveals a substantial gap between the performance of humans and MLLMs. Additionally, we explore the potential of reinforcement learning to improve visual simulation capabilities. Our findings suggest that while some models exhibit partial competence in recognizing visual patterns, robust mental visualization remains an open challenge for current MLLMs. Our dataset is publicly available at Huggingface[1], and the evaluation code can be found at GitHub[2].

## 1 Introduction

The human capacity for mental visualization – the ability to internally simulate scenes, structures, and dynamics – is central to perception and reasoning. Decades of work in cognitive science have demonstrated that people can mentally rotate objects in three-dimensional space [25], infer the future trajectory of moving bodies from a static snapshot [4], and fill in occluded or missing information in a scene based on prior experience [3]. These abilities reflect an underlying mental model of the physical and spatial world, enabling prediction in situations where direct perceptual input is limited or ambiguous.

In recent years, large language models (LLMs) have demonstrated strong performance on a wide range of linguistic and reasoning tasks, mostly driven by scale and training on massive text corpora. Building on this success, vision-language models (VLMs), which fuse textual and visual representations, have extended these capabilities to multimodal domains. VLMs have achieved impressive results on

---

[1]`https://huggingface.co/datasets/shahab7899/Hyperphantasia`
[2]`https://github.com/AIF4S/Hyperphantasia`

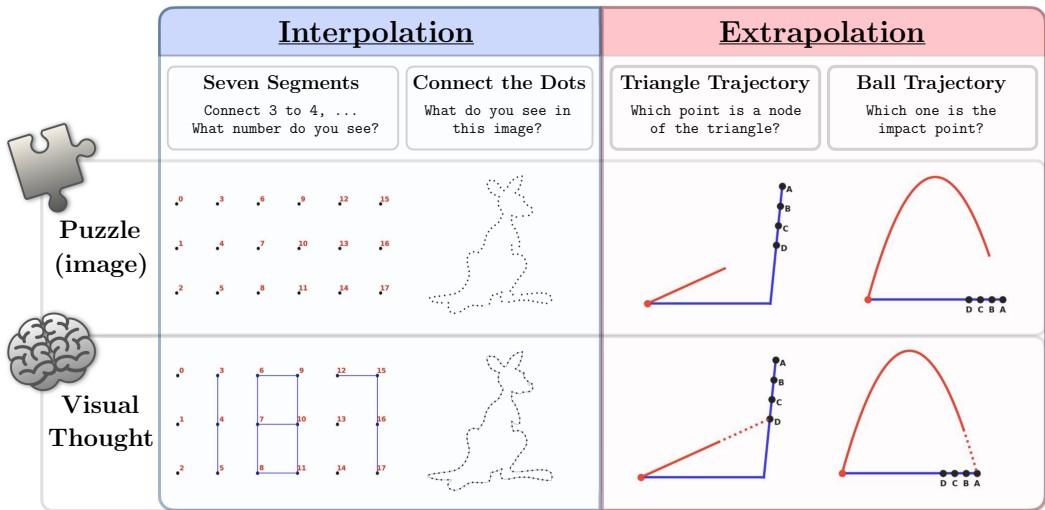

Figure 1: Overview of puzzles in Hyperphantasia

benchmarks that assess a variety of skills, including visual problem-solving [33], visual Math reasoning [19], and visual question answering (VQA) [9]. These benchmarks have become de facto standards for evaluating VLM performance. However, the majority of these tasks involve directly using the information given in the image, offering limited insight into the models' ability to simulate, extrapolate, or reason about the latent structure of a scene.

In this work, we introduce Hyperphantasia, a new benchmark suite aimed at evaluating vision-language models on tasks that require mental visualization. Our contributions are as follows:

- We present a novel synthetic benchmark consisting of four distinct puzzle types, each spanning three difficulty levels, designed to probe the mental visualization capabilities of MLLMs. The full dataset contains 1200 samples, 100 per difficulty level for each task, and is publicly available.
- We conduct a comprehensive evaluation of state-of-the-art MLLMs on our benchmark, revealing a lack of robust mental visualization abilities.
- We explore the use of reinforcement learning to elicit mental visualization behavior, and analyze how task difficulty and diversity affect model generalization and performance.
- We identify additional failure modes, beyond the lack of mental visualization, that contribute to model failure on Hyperphantasia tasks.

## 2 Background

**Vision Language Models (VLMs) –** Vision-language models (VLMs) extend the capabilities of traditional language models by incorporating visual information, enabling joint reasoning over text and images within a single architecture. Initial approaches like CLIP [21], and ALIGN [12] used contrastive learning to align visual and textual embeddings, laying the foundation for multimodal understanding. This was followed by autoregressive and encoder-decoder architectures such as Flamingo [1] and BLIP-2 [17], which introduced mechanisms to condition language generation on visual inputs through cross-attention and intermediate adapters. LLaVA [18] directly aligns a pretrained and frozen visual encoder with a language model by optimizing a projection module using image-text pairs.

**VLM Benchmarks –** The rapid development of vision-language models has been accompanied by a proliferation of benchmarks aimed at evaluating their capabilities across a range of tasks and reasoning demands. VQAv2 [9] tests models on answering natural language questions about images. To probe deeper reasoning abilities, recent benchmarks have introduced more diverse and challenging tasks. MediConfusion [22] challenges medical VLMs by introducing visually confusing image

pairs and posing differentiable questions that require fine-grained visual understanding. MMMU [33] assesses VLM performance across multiple professional domains, including medicine, law, and engineering, requiring expert-level knowledge and MathVista [19] specifically targets visual mathematical reasoning. MLLM-CompBench [13] focuses on evaluating the comparative reasoning capabilities of VLMs when given an image pair. OCRBench [7] isolates and evaluates optical character recognition (OCR) capabilities of VLMs. Other recent efforts, such as SEED-Bench [15], MM-Vet [32], and MME [6] expand this landscape by stress testing knowledge, integration of various multimodal capability combinations, and reasoning. VHELM [14] extends the HELM framework to VLMs, offering a unified evaluation across many aspects, including visual perception, knowledge, reasoning, and multilinguality, which combines/uses earlier benchmarks mentioned earlier.

There has also been a flurry of works in evaluating visual manipulation, specifically, a model's ability to mentally simulate or infer structure from incomplete visual information. Xu et al. [31] probe mental rotation and spatial manipulation of 3D objects, while SRBench [26] targets spatial reasoning of VLMs. LEGO-Puzzles [27] evaluate spatial understanding and sequential reasoning through LEGO-based tasks. While these benchmarks push toward more cognitively demanding evaluations, none systematically assess the core and extent of a model's ability to visually construct, simulate, and extrapolate information. Moreover, they do not explicitly analyze or isolate the generalization and robustness of mental visualization itself.

**Mental Visualization –** The ability to internally simulate spatial, physical, or causal properties of the world has long been studied as a core component of human cognition. Foundational experiments in cognitive psychology have demonstrated that humans can perform sophisticated internal operations on visual representations without external stimuli. Shepard and Metzler [25] showed that the time required to judge whether two 3D objects are congruent is linearly related to the angle of rotation, suggesting that people mentally rotate objects in a continuous, analog fashion. Similarly, the classic paper-folding or hole-punch experiments [28] revealed that participants could anticipate the resulting patterns of punched holes on folded paper, requiring mental transformation and spatial projection. On another thread, studies of intuitive physics have explored how people simulate object dynamics in the absence of direct observation, such as predicting the trajectory of a ball under gravity of reasoning about pulley systems and mechanical interactions [11]. These capabilities rely on the construction and manipulation of internal models of space, force, and causality.

## 3 Hyperphantasia Dataset

Our goal is to evaluate the mental visualization capabilities of Multimodal Large Language Models (MLLMs). In this context, mental visualization refers to the cognitive process of internally constructing and manipulating visual representations without explicit external stimuli. While the majority of existing benchmarks [8, 29] primarily target perception, captioning, or retrieval abilities, they offer limited insight into a model's capacity for active visual reasoning and imagination abilities that are crucial in real-world scenarios requiring dynamic visual understanding. For instance, an autonomous vehicle must be able to anticipate the trajectory of nearby moving objects to make safe and timely decisions. To address this gap, we introduce Hyperphantasia, a benchmark consisting of four synthetic tasks specifically designed to probe different aspects of mental visualization. In this section, we describe the overall design of tasks in Hyperphantasia and outline the unique cognitive abilities targeted by each task.

### 3.1 Overview of Hyperphantasia

The tasks in Hyperphantasia are organized into two main categories: Interpolation and Extrapolation, each comprising of two sub-categories. The Interpolation tasks evaluate a model's ability to infer internal boundaries and recognize visual concepts from partial information. The Extrapolation tasks assess the model's capacity to anticipate and extend visual structures beyond the information explicitly provided. In the following sections, we describe each category along with the corresponding puzzles. Figure 1 illustrates examples of each puzzle in our benchmark.

To systematically assess model performance, Hyperphantasia has three levels of difficulty: Easy, Medium, and Hard, with each level containing 100 puzzles per sub-category, resulting in a total of 1,200 samples. These difficulty levels control the visual and cognitive complexity of the tasks. Easy puzzles are straightforward and require minimal visual inference, while Medium and Hard puzzles

demand more extensive reasoning and mental simulation. Table 1 summarizes the hyperparameters used to define difficulty across tasks. Examples of puzzles at different difficulty levels are shown in Figure 2.

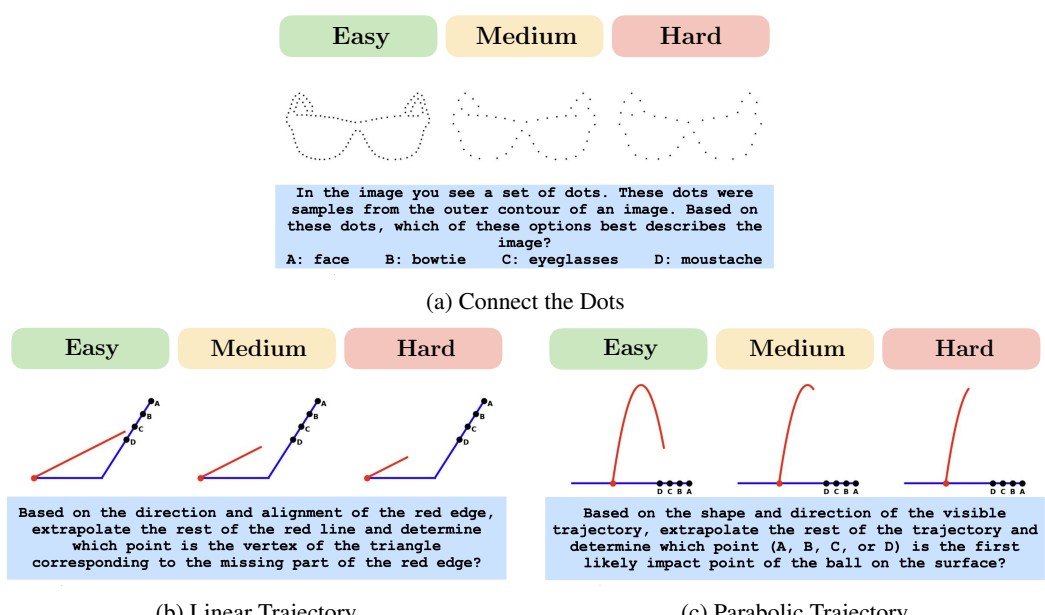

Figure 2: Examples of Hyperphantasia puzzles across difficulty levels. **(a)** In Connect the Dots, models must identify an object among four options based on samples of its outer contour. **(b)** In Linear Trajectory, the model must mentally complete a missing edge of a triangle and identify the correct vertex. **(c)** In Parabolic Trajectory, the model is asked to extrapolate the partial trajectory of a dropped ball and predict its impact point. We omit Seven Segments as it consists of uniformly styled dot grids and differs only in the number of digits per puzzle.

## 3.2 Interpolation Tasks

In these tasks, we present the model with an incomplete image and require it to infer or mentally complete the missing parts based on visual cues or explicit instructions. The goal is to assess the model's ability to interpolate missing visual information by leveraging spatial reasoning and partial patterns. The two puzzles in this category are:

- **Seven Segments**: In this puzzle, the model is shown a grid of numbered dots along with a list of edges to connect. Based on the resulting shape formed by these connections, the model must identify the digit that is implicitly drawn. The number of digits in the final shape influences the puzzle's difficulty. To generate these questions, we randomly sample a number and convert it into a set of segment-like connections inspired by seven-segment display patterns.

- **Connect the Dots**: In this task, the model is given a set of dots sampled from the external contour of an object and is asked to identify the object from four given options. To generate these puzzles, we begin by manually selecting 100 expressive images from distinct categories within the Clipart subset of the DomainNet dataset [20]. We then apply automated visual processing to extract the outer contour and sample dot points using a tunable minimum distance parameter. Larger minimum distances reduce visual fidelity and increase task difficulty. To generate the answer options, we use GPT-4o to select three visually similar but incorrect labels from the full label set and manually verify that there is only one correct option.

### 3.3 Extrapolation Tasks

This category evaluates the model's ability to predict object trajectories by mentally extending observed visual patterns, based on either linear or non-linear assumptions. Tasks in this group require the model to continue a trajectory beyond the given information in order to identify its point of impact with a specific boundary or structure. The puzzles included in this category are:

- **Linear Trajectory**: In this puzzle, the model is shown a triangle in which a portion of one edge has been removed. The task is to identify the node corresponding to the missing segment from four given points. We control the difficulty by varying how much of the edge is removed, as shorter visible segments require more precise extrapolation. To generate these puzzles, we randomly sample parameters such as the angles between edges to create a diverse set of triangle configurations.

- **Parabolic Trajectory**: These puzzles present a partial trajectory of a thrown ball, and the model is asked to determine its impact point on a horizontal surface from four options. As with the previous puzzle, difficulty is modulated by adjusting the length of the visible trajectory. Puzzles with shorter arcs demand more advanced motion prediction capabilities. Puzzle generation of this task involves random sampling of parameters such as the angle of the drop and initial height.

Table 1: Hyperparameters for different difficulty settings of Hyperphantasia.

| | Interpolation | | Extrapolation | |
|---|---|---|---|---|
| | **Seven Segments** | **Connect the Dots** | **Linear Trajectory** | **Parabolic Trajectory** |
| **Difficulty** | No. digits | Min. distance (pixel) | Visible portion of the edge (%) | Visible portion of the trajectory (%) |
| Easy | 3 | 10 | 90 | 90 |
| Medium | 4 | 20 | 60 | 60 |
| Hard | 5 | 25 | 40 | 40 |

## 4 Experiments

In this section, we describe our experimental setup and present a comprehensive evaluation of state-of-the-art Multimodal Large Language Models (MLLMs). Our results highlight the poor performance of current models on mental visualization tasks. We further analyze the distinct abilities exhibited by different models, revealing task-specific strengths and weaknesses across interpolation and extrapolation challenges. To address these limitations, we explore the use of reinforcement learning to improve performance and examine its effectiveness across puzzle types. Finally, we investigate whether augmenting input images with visual cues can help improve the models.

### 4.1 Experimental Setup

**Models:** We evaluate Hyperphantasia on a range of state-of-the-art MLLMs, including o4-mini (2025-04-16), GPT-4o (2024-08-06), Gemini 2.5 Pro (June 2025 update), Claude 3.7 Sonnet (20250219), Qwen-VL-2.5 (7B and 32B) [2], LLaMA 3.2 (11B and 90B) [10], LLaVA-OneVision (7B and 72B) [16], Molmo (7B and 72B) [5], and Deepseek-VL2 [30]. To evaluate large open-source models (above 11B), we use four NVIDIA H100 GPUs; for Deepseek-VL 2, we use three NVIDIA A100 GPUs; and for the remaining open-source models, we use a single A100 GPU. For proprietary models, we use their APIs, which do not require any significant compute resources. Evaluation time ranges from 1 to 3 hours, depending on the model.

**Human Evaluation**: For Seven Segments, each digit is clearly recognizable based on its edges, making the task trivial for humans. Moreover, we manually evaluated several Seven Segments puzzles and found them consistently easy to solve, so we report $100\%$ accuracy on this task. For the remaining puzzles, we recruited human participants while ensuring that each question was answered by at least three individuals. We report the average human accuracy across all tasks and difficulty levels. For Connect the Dots problems, since all difficulty variants are derived from the same base examples, we assigned different participants to different difficulty levels to avoid biases.

**Evaluation Protocol:** In the prompt, we allow models to explain their reasoning but require them to enclose the final answer between two <ANSWER> tags. We set the temperature to $0$ to ensure deterministic outputs and regenerate the response up to three times if the model fails to follow the required format. All evaluated models consistently adhered to this format, except for Deepseek-VL2, for which we allowed free-form output and extracted the answer via response parsing. Each puzzle includes a single image in jpg format with a resolution of $384 \times 384$ pixels.

## 4.2 Results

Table 2 reports the accuracy of various models on Hyperphantasia across the Easy, Medium, and Hard difficulty levels. Overall, current state-of-the-art MLLMs exhibit clear limitations in mental visualization, with performance degrading significantly as task difficulty increases. Additionally, we observe substantial variation in model performance across different puzzle types, indicating that models possess uneven capabilities across different tasks.

Most strikingly, all models fail on Seven Segments puzzles except for Gemini and o4-mini, which achieve $51\%$ and $83\%$ accuracy on the easy set, respectively. However, the rest of the models almost completely fail, regardless of difficulty. Upon inspecting model responses, we observe that some models repeatedly output fixed guesses regardless of the input. For example, Qwen-VL 2.5 7B answered "012" in 32 out of 100 Easy examples. This behavior suggests that these models are not engaging with the underlying visual reasoning task at all and instead resort to memorized patterns or default completions, further underscoring their lack of mental visualization ability. Examples of this behavior are provided in the Appendix 8.

Unlike Seven Segments puzzles, models perform relatively well on Connect the Dots puzzles in the Easy setting, with Gemini and GPT-4o reaching $97\%$ and $96\%$, respectively. However, performance degrades significantly on Medium and Hard, with Gemini being the best model with $75\%$ accuracy on the Hard set. While human accuracy also declines with increased difficulty, the drop is much smaller (around $4\%$) and the accuracy consistently remains above $94\%$. Furthermore, despite the relative simplicity of the task, models such as Molmo and Deepseek struggle with Medium and Hard sets and perform near the level of random guessing. Upon closer inspection of model outputs, we find that some models confidently justify incorrect answers by hallucinating features related to the wrong option, which are not present in the image. This suggests that, rather than grounding their answers in the visual input, models often rely on loosely associated or imagined patterns, revealing limitations in both visual grounding and fine-grained perceptual reasoning. We provide examples of such answers in the Appendix 8.

Extrapolation tasks also prove to pose a major challenge to the models. In Linear Trajectory puzzles, Claude achieves $60\%$ in the Easy set, the highest among all models, but drops to the level of random guessing on Medium and Hard. o4-mini shows a similar trend, scoring $43\%$ on Easy but struggling with more difficult examples. Interestingly, LLaVA-OneVision 72B, which performs modestly on Connect the Dots puzzles, performs on par with 4o-mini on Linear Trajectory Easy difficulty level. Overall, except for Claude, all models have below $50\%$ accuracy in Easy Linear Trajectory puzzles. The struggle with this task is particularly notable given that the human accuracy on Easy problems is $100\%$. In this puzzle, we see a larger drop in human accuracy as the difficulty level increases, but the accuracy remains fairly high ($89\%$ on Hard).

Parabolic Trajectory puzzles are even more difficult for the models. Although Claude again has the best performance in the Easy set, its accuracy rapidly deteriorates in the Medium and Hard sets. Moreover, almost all models fall to near random guessing on the Medium and Hard sets. One exception is LLaMA 3.2 72B, which achieves $31\%$ in Medium Parabolic Trajectory despite underperforming on earlier tasks. LLaVA-OneVision continues to perform competitively, again on par with o4-mini and outperforming the remaining models. A noteworthy trend we observe in responses to extrapolation tasks is that some models resort to overly simplistic and incorrect heuristics, such as selecting the rightmost point as the impact location. This further demonstrates a failure to perform genuine trajectory reasoning. Examples of this behavior are provided in Appendix 8. Notably, human accuracy on Easy examples is close to $100\%$, but drops substantially to around $50\%$ on Medium and Hard. We hypothesize that participants can eliminate two implausible options but often struggle to confidently choose between the final two, leading to a near-random guessing performance.

Overall, these findings underscore the limitations of current state-of-the-art MLLMs in performing mental visualization tasks, revealing a significant gap between models and human capabilities. Even the best-performing models struggle with tasks that require mentally constructing or extending visual structures, which are intuitive and nearly trivial for humans. This persistent gap emphasizes the need for deeper investigation into how MLLMs can acquire the ability to reason over visual abstractions.

Table 2: Accuracy (%) of models on Hyperphantasia. We report the best accuracy for each puzzle across difficulties in **bold** and underscore the second-best accuracy.

| | Interpolation | | | | | | Extrapolation | | | | | | Mean | | |
| | Seven Segments | | | Connect the Dots | | | Linear Trajectory | | | Parabolic Trajectory | | | | | |
| Model | Easy | Medium | Hard | Easy | Medium | Hard | Easy | Medium | Hard | Easy | Medium | Hard | Easy | Medium | Hard |
|---|---|---|---|---|---|---|---|---|---|---|---|---|---|---|---|
| o4-mini | **83** | **85** | **85** | 90 | 69 | 64 | 43 | 26 | 23 | 35 | 25 | **33** | 62.75 | 51.25 | 51.25 |
| GPT4-o | 3 | 4 | 0 | 96 | **80** | 59 | 28 | 23 | 24 | 16 | 17 | 27 | 35.75 | 31.00 | 27.50 |
| Gemini 2.5 pro | 51 | 44 | 40 | **97** | 74 | **75** | 31 | 26 | **29** | 26 | 24 | 23 | 51.25 | 42.00 | 41.75 |
| Claude 3.7 Sonnet | 1 | 0 | 0 | 86 | 56 | 49 | **60** | 19 | 24 | **40** | 21 | 27 | 44.25 | 24.00 | 25.00 |
| Qwen VL 2.5 7B | 0 | 0 | 0 | 66 | 36 | 35 | 27 | 18 | 24 | 16 | 22 | 18 | 27.25 | 19.00 | 19.25 |
| Qwen VL 2.5 32B | 1 | 0 | 0 | 68 | 32 | 34 | 40 | 24 | 27 | 19 | 23 | 25 | 32.00 | 19.75 | 21.50 |
| Llama 3.2 11B | 0 | 0 | 0 | 64 | 39 | 28 | 36 | 18 | 24 | 32 | 26 | 22 | 33.00 | 20.75 | 18.50 |
| Llama 3.2 90B | 0 | 0 | 0 | 83 | 40 | 43 | 30 | 23 | 22 | 28 | 31 | 26 | 35.25 | 23.50 | 22.75 |
| LLaVA-OneVision 7B | 0 | 0 | 0 | 92 | 64 | 52 | 22 | **29** | 27 | 19 | 28 | 22 | 33.25 | 30.48 | 25.25 |
| LLaVA-OneVision 72B | 0 | 0 | 0 | 89 | 44 | 43 | 42 | 27 | 26 | 32 | **34** | 22 | 40.75 | 26.25 | 22.75 |
| Molmo 7B | 0 | 0 | 0 | 66 | 28 | 28 | 28 | 19 | 20 | 27 | 19 | 20 | 30.25 | 16.50 | 17.00 |
| Molmo 72B | 0 | 0 | 0 | 62 | 32 | 24 | 29 | 27 | 25 | 26 | 29 | 22 | 29.25 | 22.00 | 17.75 |
| Deepseek-VL2 | 0 | 0 | 0 | 75 | 38 | 28 | 20 | 11 | 11 | 12 | 15 | 14 | 26.75 | 16.00 | 13.00 |
| Human | 100.00 | 100.00 | 100.00 | 98.86 | 94.00 | 95.20 | 100.00 | 91.33 | 89.33 | 100.00 | 54.40 | 52.33 | 99.72 | 84.93 | 84.22 |
| Random Guess | 0.00 | 0.00 | 0.00 | 25.00 | 25.00 | 25.00 | 25.00 | 25.00 | 25.00 | 25.00 | 25.00 | 25.00 | 25.00 | 25.00 | 25.00 |

## 4.3 Improving Mental Visualization via Reinforcement Learning

We investigate Reinforcement Learning (RL) as a potential remedy for the poor performance of MLLMs on mental visualization tasks in Hyperphantasia. We focus on RL rather than Supervised Fine-Tuning (SFT), as our goal is to improve the model's ability to reason about the problem. However, we do not have access to detailed thinking traces, and without such supervision, SFT typically fails to generalize from easier examples to harder ones.

Due to the limited number of Connect the Dots samples and the fact that models can solve Seven Segments puzzles by memorizing edge patterns, bypassing the need for genuine visual reasoning, we restrict our RL experiments to the extrapolation tasks. We construct multiple training datasets with varying difficulty and task composition, as summarized in Table 3. For each experiment, we generate new training and test samples and train Qwen-VL 2.5 7B using the publicly available GRPO [23] implementation from Sheng et al. [24]. We select the best model checkpoint based on test loss and evaluate it on the original extrapolation puzzles of Hyperphantasia.

We use four NVIDIA H100 GPUs for training and follow the implementation and hyperparameter settings provided by Sheng et al. [24]. Models are trained for 30, 20, and 15 epochs for datasets containing 2000, 3000, and 4000 samples, respectively. Each training run takes roughly 7 hours.

Table 4 reports the accuracy of trained models across Hyperphantasia and their respective test samples. We find that training solely on Easy puzzles results in weak generalization to more difficult puzzles. For instance, training the model on Easy Linear Trajectory puzzles causes it to adopt a superficial heuristic of selecting the point closest to the red line which works for the Easy set, but results in an extremely poor performance on medium Linear Trajectory puzzles, yielding only 9% accuracy, well below random guessing. In contrast, models trained on Medium puzzles exhibit stronger generalization. For instance, the model trained on Medium Parabolic Trajectory puzzles surpasses o4-mini and achieves 40% accuracy on Hard Parabolic Trajectory, a substantial gain given that human accuracy is 52.33% and the model only has 7B parameters.

Our experiments exhibit that combining Easy and Medium puzzles further improves generalization. The model trained on mixed Parabolic Trajectory puzzles attains high performance on different difficulties of Parabolic Trajectory puzzles with 44% accuracy on Hard Parabolic Trajectory, which is much better than o4-mini. This model also achieves the accuracy of 44% in Easy Linear Trajectory puzzles, suggesting cross-task transfer of mental visualization. Among broader mixtures, the model

trained on the All Mix dataset performs well across tasks but appears to overfit to easier Linear Trajectory examples. In contrast, the model trained on the Hard Mix dataset, which does not include Easy Linear Trajectory examples, demonstrates robust and consistent performance across all tasks.

These findings suggest that training on Easy examples alone is not only insufficient but may also encourage reliance on shallow heuristics. However, training on diverse and non-trivial problems enables models to develop more generalized and transferable mental visualization capabilities.

Table 3: Datasets used for RL training.

| | Training Samples | | | | | Test Samples | | | | |
| | Linear Trajectory | | Parabolic Trajectory | | | Linear Trajectory | | Parabolic Trajectory | | |
| Name | Easy | Medium | Easy | Medium | Total | Easy | Medium | Easy | Medium | Total |
|---|---|---|---|---|---|---|---|---|---|---|
| Linear Easy | 2000 | 0 | 0 | 0 | 2000 | 300 | 0 | 0 | 0 | 300 |
| Linear Medium | 0 | 2000 | 0 | 0 | 2000 | 0 | 300 | 0 | 0 | 300 |
| Parabolic Easy | 2000 | 0 | 0 | 0 | 2000 | 300 | 0 | 0 | 0 | 300 |
| Parabolic Medium | 0 | 0 | 0 | 2000 | 2000 | 0 | 0 | 0 | 300 | 300 |
| Linear Mix | 1000 | 1000 | 0 | 0 | 2000 | 150 | 150 | 0 | 0 | 300 |
| Parabolic Mix | 0 | 0 | 1000 | 1000 | 2000 | 0 | 0 | 150 | 150 | 300 |
| All Mix | 1000 | 1000 | 1000 | 1000 | 4000 | 150 | 150 | 150 | 150 | 600 |
| Hard Mix | 0 | 1000 | 1000 | 1000 | 3000 | 0 | 200 | 200 | 200 | 600 |

Table 4: Accuracy (%) of models trained with reinforcement learning on the Extrapolation puzzles of Hyperphantasia.

| | | Test Data | | | | | |
| | | Linear Trajectory | | | Parabolic Trajectory | | |
| Training Data | Test Samples | Easy | Medium | Hard | Easy | Medium | Hard |
|---|---|---|---|---|---|---|---|
| None (Base model) | - | 27 | 18 | 24 | 16 | 22 | 18 |
| Linear Easy | 98.33 | 95 | 9 | 24 | 21 | 18 | 19 |
| Linear Medium | 51.33 | 50 | 52 | 30 | 22 | 18 | 18 |
| Parabolic Easy | 65.33 | 28 | 17 | 22 | 36 | 26 | 22 |
| Parabolic Medium | 50.67 | 25 | 23 | 29 | 18 | 39 | 40 |
| Linear Mix | 63.67 | 85 | 36 | 37 | 21 | 21 | 21 |
| Parabolic Mix | 63.00 | 44 | 23 | 25 | 60 | 52 | 44 |
| All Mix | 46.50 | 74 | 35 | 35 | 30 | 33 | 29 |
| Hard Mix | 46.17 | 37 | 36 | 32 | 35 | 35 | 32 |

## 4.4 Investigating Model Failure on Seven Segments Puzzles

To better understand the surprisingly poor performance of models on Seven Segments puzzles, we conduct an additional experiment in which we augment the input images of the Easy set by explicitly drawing the edges between the specified points. An example of such a modified image is shown in Figure 3. The original prompt remains unchanged, but the added edges serve as clear visual cues that should make the intended digit immediately recognizable.

Table 5 summarizes the results of this experiment. Strikingly, only GPT-4o and Gemini are able to leverage the visual cues and solve the task, achieving 76% and 98% accuracy, respectively. All other models continue to fail, with 0% accuracy across the board. These results suggest that, at least for Seven Segments puzzles, the failure of most models cannot be fully attributed to the lack of mental visualization alone. Instead, most models appear to struggle with extracting and interpreting even simple geometric patterns when presented in a slightly out-of-distribution visual format that is otherwise trivial for humans.

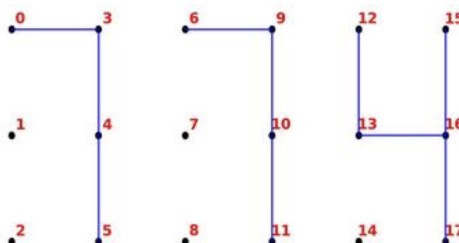

Figure 3: An example of Seven Segments puzzle with added edges as visual cues.

Table 5: Accuracy of models on Seven Segments puzzles when the edges are explicitly added to the image.

| Model | Accuracy (%) |
|---|---|
| GPT-4o | 76 |
| Gemini 2.5 Pro | 98 |
| Claude 3.7 Sonnet | 0 |
| Qwen VL2.5 7B | 0 |
| Llama 3.2 11B | 0 |
| LLaVA-OneVision 7B | 0 |

## 5 Discussion

### 5.1 Mental Visualization in MLLMs

Our results indicate that current MLLMs lack consistent and generalizable mental visualization capabilities. Performance is highly task-dependent: some models perform well on extrapolation tasks but fail on interpolation tasks, and vice versa. This inconsistency suggests that models do not possess a unified internal mechanism for visual reasoning, but instead rely on task-specific shortcuts. Furthermore, we observe cases where models hallucinate visual features that are not present in the image, falsely grounding their predictions in imagined evidence. This behavior implies that, rather than genuinely engaging in visual reasoning, models may be mimicking it through learned patterns, without a true understanding of the underlying visual content.

Moreover, model performance often deteriorates sharply with even modest increases in task difficulty. This is particularly evident in the steep drop from Easy to Medium levels in tasks such as Connect the Dots or Linear Trajectory. While these puzzles remain simple for humans despite added complexity, most models fail to adapt, revealing the brittleness of their mental visualization capabilities. Unlike human cognition, which can rely on approximate or partial inferences in the face of uncertainty, MLLMs tend to break down entirely, lacking the flexibility to generalize beyond narrowly defined settings.

### 5.2 Learning Mental Visualization via Reinforcement Learning

Our RL experiments demonstrate that mental visualization abilities can be learned through training, even with a relatively small model, provided that the training signal is carefully structured. Exposure to diverse and moderately challenging tasks enables models to generalize to harder and even novel tasks. In contrast, training on overly simplistic datasets with limited variability encourages brittle heuristics that fail under minor distribution shifts. These findings underscore that improving mental visualization is not solely a matter of scale or capacity, but it depends critically on training data that is both diverse and sufficiently challenging to promote robust and transferable skills.

### 5.3 Visual Recognition Failures in Seven Segments

Our targeted intervention in Seven Segments tasks by adding explicit visual cues revealed a surprising failure mode. Despite removing the need for internal construction, most models still failed to interpret the rendered image. This suggests that poor performance is not solely due to weak mental visualization but also reflects an inability to extract structure from slightly out-of-distribution visual inputs. The failure to recognize even clean, simple geometric shapes under minor format shifts highlights a broader limitation in visual robustness that undermines performance across many tasks.

These results point to a missing layer in current MLLMs, the ability to do visual reasoning over structured visual abstractions in a robust and flexible manner. Without this capability, models remain vulnerable to small visual perturbations and fail to generalize beyond tightly constrained and simple visual tasks.

### 5.4 Future Directions

Beyond reinforcement techniques, we believe that a promising direction is to equip models with visual thinking capabilities. Current models reason entirely in the language domain, but Hyperphantasia puzzles and many real-world scenarios, the thinking cannot be explained with language, and they require visual thinking. While recent "omni" models offer some potential in this area, they are still significantly behind state-of-the-art language models in overall performance.

Importantly, we do not believe that omni models are the only path forward. Visual thinking tokens do not need to correspond to meaningful or interpretable images. Instead, they could function as internal visual representations. However, designing and training such models requires careful consideration, especially in how to cue or supervise this form of internal visualization. Some early efforts in this direction, such as [1], have explored visual reasoning in constrained setups, but these remain narrow in scope. We see Hyperphantasia as an ideal testbed to encourage progress in this direction.

## 6    Conclusion

In this work, we introduce Hyperphantasia, a novel synthetically generated benchmark designed to evaluate the mental visualization capabilities of MLLMs. The dataset comprises four puzzle types, each spanning three difficulty levels and organized into two categories. Our evaluation of various open-source and proprietary state-of-the-art MLLMs reveals that existing models lack consistent and generalizable mental visualization abilities. To remedy this issue, we explore the use of reinforcement learning to elicit mental visualization and find that, when trained on moderately difficult and diverse examples, models can begin to generalize to more difficult and even new tasks. Additionally, our analysis highlights another key limitation: models often fail to interpret slightly out-of-distribution visual inputs, even when the task is visually simple. Together, these findings position Hyperphantasia as an effective benchmark for measuring and developing visual reasoning capabilities in multimodal models. However, we note that Hyperphantasia focuses on a small set of tasks and thus captures only a subset of the broader space of mental visualization capabilities. Future work may expand the benchmark with additional task variety to more comprehensively evaluate visual imagination in MLLMs.

## Acknowledgements

We would like to thank Microsoft for an Accelerating Foundation Models Research grant that provided the OpenAI credits enabling this work. This research is also in part supported by AWS credits through an Amazon Faculty research award and a NAIRR Pilot award. M. Soltanolkotabi is also supported by the Packard Fellowship in Science and Engineering, a Sloan Research Fellowship in Mathematics, an NSF-CAREER under award #1846369, DARPA FastNICS program, and NSF-CIF awards #1813877 and #2008443. and NIH DP2LM014564-01.

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

# Appendix

## 7 Evaluation Prompts

In this part, we provide the exact prompts used for the evaluation of each task.

### 7.1 Seven Segments

For these puzzles, the edges to connect and the number of digits vary among puzzles, but the rest of the prompt is the same for all samples.

> Solve the following puzzle. The rules are as follows.
> 1. You see a grid of dots.
> 2. Every dot is denoted by a red number on top right of it.
> 3. Connect the following dots:
> **EDGES TO CONNECT**
> 4. What is the **N**-digit number formed in the image after drawing these connections?
>
> You can explain your reasoning but you should specify your final answer by putting <ANSWER> before and after it like <ANSWER>FINAL_ANSWER<ANSWER> where FINAL_ANSWER is your answer. For example, if your answer is A, you should say <ANSWER>A<ANSWER>

### 7.2 Connect the Dots

In these prompts, the options vary for different puzzles.

> In the image you see a set of dots. These dots were sampled from the outer contour of an image. Based on these dots, which of these options best describes the image?
> A: **OPTION A**
> B: **OPTION B**
> C: **OPTION C**
> D: **OPTION D**
> Answer with the letter of the option that you think is correct.
>
> You can explain your reasoning but you should specify your final answer by putting <ANSWER> before and after it like <ANSWER>FINAL_ANSWER<ANSWER> where FINAL_ANSWER is your answer. For example, if your answer is A, you should say <ANSWER>A<ANSWER>

### 7.3 Linear Trajectory

We use the same prompt for all of the puzzles.

> The image shows a triangle composed of two blue edges and one red edge. A portion of the red edge has been removed. The red line originates from the left red vertex and should connect to a point along the blue edge on the right side of the triangle. There are four possible points marked on this blue edge: A, B, C, and D. Question: Based on the direction and alignment of the red edge, extrapolate the rest of the red line and determine which point (A, B, C, or D) is the vertex of the triangle corresponding to the missing part of the red edge? Select the most geometrically consistent option.
>
> You can explain your reasoning but you should specify your final answer by putting <ANSWER> before and after it like <ANSWER>FINAL_ANSWER<ANSWER> where FINAL_ANSWER is your answer. For example, if your answer is A, you should say <ANSWER>A<ANSWER>

### 7.4 Parabolic Trajectory

We use the same prompt for all of the puzzles.

You are given an image showing part of a ball's parabolic trajectory (in red) as it moves through the air. The ball will eventually land on a horizontal surface shown in blue. Along this surface, four positions are marked: A, B, C, and D (from right to left). Question: Based on the shape and direction of the visible trajectory, extrapolate the rest of the trajectory and determine which point (A, B, C, or D) is the first likely impact point of the ball on the surface?

## 8 Examples of Hallucinations in the Models

In this part, we provide examples of hallucination and fix answers that some models provide.

### 8.1 Seven Segments

For this puzzle, some models tend to repeatedly produce a fixed answer, such as 012, regardless of changes in the input. These default responses vary across models, and there is no single answer that is universally repeated. Additionally, some of the generated answers contain a different number of digits than what is specified in the prompt, indicating a failure to follow basic task constraints. Table 6 summarizes this behavior for a subset of models that exhibit these patterns.

Table 6: Number of occurrences for the most frequent responses across levels of difficulties in Seven Segments.

| Model | Difficulty | Answer | Number of occurrence (out of 100) |
|---|---|---|---|
| | Easy | 409 | 20 |
| Claude 3.7 Sonnet | Medium | 2023 | 60 |
| | Hard | 20235 | 10 |
| | Easy | 012 | 32 |
| Qwen VL 2.5 7B | Medium | 0123 | 26 |
| | Hard | 01234 | 18 |
| | Easy | 888 | 29 |
| Qwen VL 2.5 32B | Medium | 2023 | 25 |
| | Hard | 22222 or 2023 | 17 |
| | Easy | 123 | 50 |
| LLaVA-OneVision 7B | Medium | 0123 | 25 |
| | Hard | 12345 | 22 |
| | Easy | 170 or 161 | 28 |
| Molmo 7B | Medium | 2300 or 2310 | 25 |
| | Hard | 272829 | 32 |

### 8.2 Connect the Dots

In this part, we provide examples of answers in which the model hallucinates non-existent features. **Example 1:** We showed Figure 4a to Molmo 7B, and in its response, it is hallucinating features of a swan:

**Molmo**: The image appears to show a dotted outline of a bird in flight. The shape is elongated and curved, with a distinctive neck and head form that's characteristic of certain waterfowl. The overall silhouette suggests a graceful, soaring posture.

Given the options provided, the bird that most closely matches this dotted outline is:

<ANSWER>A<ANSWER>

This dotted shape best represents a swan in flight. Swans are known for their long, elegant necks and graceful forms, which align well with the curved outline visible in the dotted image. The elongated body and the way the neck curves upwards to form the head are particularly indicative of a swan's silhouette.

**Example 2:** We showed Figure 4b to Llama 3.2 11B, and it hallucinated onion features:

**Llama**: To determine the correct answer, let's analyze the shape formed by the dots. The dots outline a teardrop shape with a narrow top and a wider bottom. This shape is most similar to an onion, which has a similar teardrop shape with a narrow top and a wider bottom.

The other options do not match the shape formed by the dots. A carrot is typically long and thin, an onion is bulbous with a narrow top and a wider bottom, and a pineapple is round with a crown on top.

Therefore, based on the shape formed by the dots, the correct answer is:

<ANSWER>B<ANSWER>"

**Example 3:** We showed Figure 4a to Claude 3.7 Sonnet, and it hallucinated features of a swan:

**Claude**: Looking at the dotted outline in the image, I can see a shape that appears to have:
- A long, curved neck
- A rounded body
- What looks like a tail at the back
- A distinctive shape at the bottom that could represent feet or a base

The overall silhouette strongly resembles a bird with a long, curved neck. Among the options:

A: Swan - Swans have long, curved necks and a distinctive body shape that matches this outline quite well
B: Flamingo - Flamingos have long necks but they're typically shown with their necks in an S-curve and standing on one leg
C: Parrot - Parrots typically have shorter necks and a more compact body shape
D: Butterfly - Butterflies have a completely different shape with wings

The curved neck, body proportions, and overall posture most closely match a swan in profile. The dots appear to outline a swan with its neck curved gracefully and its body in a resting position.

<ANSWER>A<ANSWER>

**Example 4:** We showed Figure 4c to Qwen VL 2.5 32B, and the model hallucinated the existence of features of a snail's shell.

**Qwen**: To determine which option best describes the image based on the given set of dots, let's analyze the shape formed by the dots:

1. **Shape Analysis**:
- The dots form a continuous, curved line that loops back on itself.
- The overall shape resembles a spiral or a coiled structure.

2. **Comparison with Options**:
- **A: Sea Turtle**: Sea turtles typically have a more defined shell shape with straighter edges and a distinct head and tail. The given shape does not resemble a sea turtle.
- **B: Snake**: Snakes have a long, slender body that is usually straight or slightly curved. The given shape is too coiled to represent a snake.
- **C: Snail**: Snails have a spiral shell, which matches the coiled, looping pattern of the dots in the image.
- **D: Lobster**: Lobsters have a segmented body with claws and antennae, which do not match the coiled shape of the dots.

3. **Conclusion**:
- The shape formed by the dots most closely resembles the spiral shell of a snail.

<ANSWER>C<ANSWER>

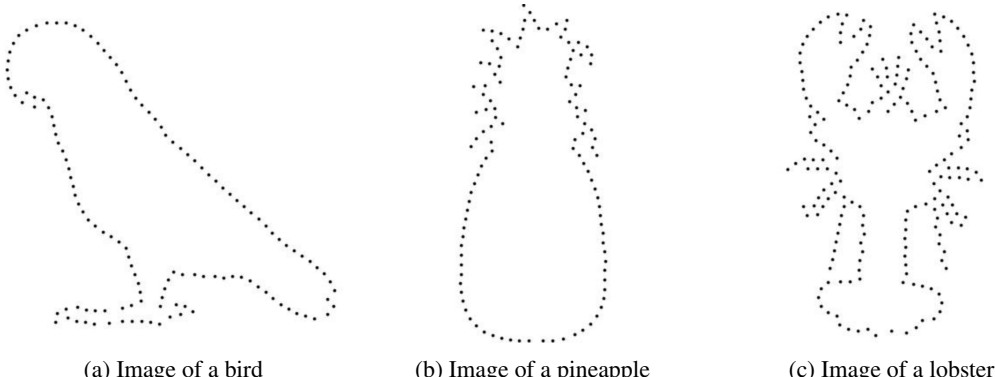

(a) Image of a bird    (b) Image of a pineapple    (c) Image of a lobster

Figure 4: Images which caused hallucination in the models.

## 8.3  Extrapolation Puzzles

We observe that some models try to find over simplistic heuristics to answer the extrapolation questions. In this part, we provide examples of this behavior.

**Example1:** After seeing Figure 5a, Qwen VL 2.5 32B thinks that the farthest point to the right should be the answer for the Parabolic Trajectory puzzle.

**Qwen:** To determine the first likely impact point of the ball on the surface, let's analyze the given information step by step:
1. **Shape of the Trajectory**: The red curve represents a parabolic trajectory, which is typical for objects moving under the influence of gravity. Parabolas are symmetric about their vertex, and the path of the ball will continue downward after reaching the peak.
2. **Direction of Motion**: The ball is moving from left to right along the parabolic path. This means that as the ball descends, it will continue moving horizontally until it intersects with the blue surface.
3. **Extrapolation of the Path**: Since the ball is already descending and moving toward the right, we can infer that the ball will continue its downward motion and intersect the blue surface at some

point. The point of intersection will be the first point where the parabolic path meets the horizontal surface.

4. **Positioning of Points A, B, C, and D**: The points A, B, C, and D are arranged from right to left along the blue surface. Given the direction of the ball's motion (from left to right), the ball will first reach the farthest point to the right on the surface.

5. **Conclusion**: Since the ball is moving from left to right and will first hit the surface at the point closest to its current position, the first likely impact point is **A**.

Thus, the final answer is:

<ANSWER>A<ANSWER>

**Example 2:** Molmo 7B thinks that the lower point should be the answer when answering Figure 5b.

**Molmo:** To determine the most geometrically consistent point for the vertex of the triangle, I'll consider the direction and alignment of the red edge:

1. The red edge starts from the left red vertex and extends diagonally upward to the right.

2. It intersects with the blue edge at four possible points: A, B, C, and D.

3. The edge should continue in a straight line after the intersection.

Given these observations, the most geometrically consistent option would be one that maintains the straight line continuation of the red edge. Points A, B, and C are all along the blue edge, which suggests they could be potential continuation points. However, point D stands out as it's positioned slightly lower on the blue edge compared to A, B, and C.

This positioning of D makes it more likely to be the continuation point that would maintain the straight line of the red edge. It creates a more balanced and symmetrical triangle compared to using A, B, or C.

<ANSWER>D<ANSWER>

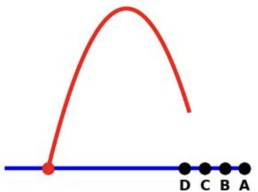

(a) An image of Parabolic Trajectory puzzle

(b) An image of Linear Trajectory puzzle

Figure 5: Extrapolation images shown to the models.

# 9 Ablation on Prompting

In this section, we explore different prompting techniques and their effects on models' performance on Hyperphantasia.

## 9.1 Investigating the prompt sensitivity of the models

To ensure fairness in comparison, we conduct an experiment with different prompt formatting to investigate the effects of prompt formatting on models' performance. For this purpose, we curated four alternative prompts and used them for answering the Easy Linear Trajectory task. Table 7 provides the curated alternative prompts, and the results are in Table 8.

Table 7: List of prompts we use to evaluate sensitivity of model performance to prompt variations. The second column shows the high-level rationale for the variation, and the specific prompt is included in the last column.

| # | Change | Prompt |
|---|--------|--------|
| 0 | Original | The image shows a triangle composed of two blue edges and one red edge. A portion of the red edge has been removed. The red line originates from the left red vertex and should connect to a point along the blue edge on the right side of the triangle. There are four possible points marked on this blue edge: A, B, C, and D. Question: Based on the direction and alignment of the red edge, extrapolate the rest of the red line and determine which point (A, B, C, or D) is the vertex of the triangle corresponding to the missing part of the red edge? Select the most geometrically consistent option. You can explain your reasoning but you should specify your final answer by putting <ANSWER> before and after it like <ANSWER>FINAL_ANSWER<ANSWER> where FINAL_ANSWER is your answer. For example, if your answer is A, you should say <ANSWER>A<ANSWER>. |
| 1 | Shorten | Based on the incomplete edge of the triangle in the image, which point among A, B, C, and D is a vertex of the triangle? You can explain your reasoning but you should specify your final answer by putting <ANSWER> before and after it like <ANSWER>FINAL_ANSWER<ANSWER> where FINAL_ANSWER is your answer. For example, if your answer is A, you should say <ANSWER>A<ANSWER>. |
| 2 | Instruct the model for extra care and attention | Carefully and skillfully review the image and answer the following question. The image shows a triangle composed of two blue edges and one red edge. A portion of the red edge has been removed. The red line originates from the left red vertex and should connect to a point along the blue edge on the right side of the triangle. There are four possible points marked on this blue edge: A, B, C, and D. Question: Based on the direction and alignment of the red edge, extrapolate the rest of the red line and determine which point (A, B, C, or D) is the vertex of the triangle corresponding to the missing part of the red edge? Carefully, select the most geometrically consistent option. You can explain your reasoning but you should specify your final answer by putting <ANSWER> before and after it like <ANSWER>FINAL_ANSWER<ANSWER> where FINAL_ANSWER is your answer. For example, if your answer is A, you should say <ANSWER>A<ANSWER>. |
| 3 | Instruct the model for single letter answering | The image shows a triangle composed of two blue edges and one red edge. A portion of the red edge has been removed. The red line originates from the left red vertex and should connect to a point along the blue edge on the right side of the triangle. There are four possible points marked on this blue edge: A, B, C, and D. Question: Based on the direction and alignment of the red edge, extrapolate the rest of the red line and determine which point (A, B, C, or D) is the vertex of the triangle corresponding to the missing part of the red edge? Select the most geometrically consistent option. Your answer should be a single letter: A, B, C, or D without any explanation or additional text. |
| 4 | Define AI expert | You are an expert mental visualization AI with deep visual thinking capabilities. The image shows a triangle composed of two blue edges and one red edge. A portion of the red edge has been removed. The red line originates from the left red vertex and should connect to a point along the blue edge on the right side of the triangle. There are four possible points marked on this blue edge: A, B, C, and D. Question: Based on the direction and alignment of the red edge, extrapolate the rest of the red line and determine which point (A, B, C, or D) is the vertex of the triangle corresponding to the missing part of the red edge? Select the most geometrically consistent option. You can explain your reasoning but you should specify your final answer by putting <ANSWER> before and after it like <ANSWER>FINAL_ANSWER<ANSWER> where FINAL_ANSWER is your answer. For example, if your answer is A, you should say <ANSWER>A<ANSWER>. |

Table 8: Results of prompt variations.

| Model | Prompt 0 | Prompt 1 | Prompt 2 | Prompt 3 | Prompt 4 |
|---|---|---|---|---|---|
| Qwen VL 2.5 7B | 27 | 32 | 26 | 23 | 25 |
| Qwen VL 2.5 32B | 40 | 38 | 35 | 35 | 27 |
| Llama 3.2 11B | 36 | 43 | 39 | 27 | 41 |
| Llama 3.2 90B | 30 | 28 | 32 | 21 | 27 |
| LLaVA-OneVision 7B | 26 | 40 | 32 | 41 | 24 |
| LLaVA-OneVision 72B | 42 | 43 | 26 | 39 | 34 |

These results highlight the fact that the prompt formatting has little effect on the performance of the models and it cannot solve the underlying problem.

## 9.2 Multi-shot prompting

To further explore the effects of prompting, we conducted an additional experiment with 3-shot prompting. Similar to the previous setup, we use the Easy Linear Trajectory puzzles for this experiment. Table 9 shows the results of this experiment. We do not report the 3-shot accuracy of LLaVA-OneVision 72B, as it was unable to produce a meaningful answer when multiple images were included in the prompt. We speculate that this may be due to the lack of multi-image inputs in its training data. Interestingly, this issue does not arise with the smaller version of the model, which was able to provide valid answers.

Table 9: Results of multi-shot prompting.

| Model | Zero-Shot | 3-Shot |
|---|---|---|
| o4-mini | 43 | 40 |
| Qwen VL 2.5 7B | 27 | 30 |
| Qwen VL 2.5 32B | 40 | 31 |
| LLaVA-OneVision 7B | 26 | 29 |
| LLaVA-OneVision 72B | 42 | - |
| Llama 3.2 11B | 36 | 35 |
| Llama 3.2 90B | 30 | 31 |

As with the previous experiment, we observe minimal to no improvement in model performance, suggesting that multi-shot prompting alone is insufficient to overcome the fundamental limitations of current models in mental visualization tasks.

## 10 Language-only Seven Segments task

Seven Segments puzzles are the only task in Hyperphantasia that can be completely explained in a language-only format without showing the image. As a result, we curate a new text-only prompt for the Easy Seven Segments puzzles and use it to test some of the previously tested models and language-only models. Here is the curated prompt:

Solve the following puzzle:
Imagine a 3 by 6 grid of dots arranged as follows (rows: top to bottom, columns: left to right):

Column 1: [0] [1] [2]
Column 2: [3] [4] [5]
Column 3: [6] [7] [8]
Column 4: [9] [10] [11]
Column 5: [12] [13] [14]
Column 6: [15] [16] [17]

Now, imagine the following pairs of dots are connected with straight lines:
<CONNECTIONS>

What 3-digit number is formed by these connections?

In the prompt, <CONNECTIONS> refers to the specific connections of each puzzle. Table 10 summarizes the results of this experiment.

Table 10: Results of the text-only Seven Segments task. The second column indicates whether the model is Language-only (L) or Vision-Language (VL). The final column reports the accuracy of each Vision-Language model on the original image-based version of this puzzle.

| Model | Model type | Accuracy in text-only format | Accuracy in image format |
|---|---|---|---|
| o4-mini | VL | 88 | 83 |
| GPT-4o | VL | 1 | 4 |
| Qwen VL 2.5 7B | VL | 0 | 0 |
| Qwen VL 2.5 32B | VL | 1 | 0 |
| Qwen 2.5 32B Instruct | L | 0 | - |
| DeepSeek-R1 distill Qwen 32B | L | 0 | - |
| Llama 3 8B Instruct | L | 0 | - |

These results demonstrate that language-only models are unable to solve this task. This outcome is expected, as these models are not trained on any visual data and therefore lack the mental visualization capabilities. Additionally, we observe that the multi-modal models which previously succeeded on this task (such as o4-mini and, to a lesser extent, GPT-4o) continue to solve it even in the text-only format. Meanwhile, other models that initially failed still struggle with this version of the task.

