# OpenReview forum: "Hyperphantasia: A Benchmark for Evaluating the Mental Visualization Capabilities of Multimodal LLMs"
_NeurIPS.cc/2025/Datasets_and_Benchmarks_Track — NeurIPS 2025 Datasets and Benchmarks Track poster_

### Official Review · Reviewer_cEcQ · 2025-06-06

**Rating:** 4
**Confidence:** 4

**Summary:**

This paper introduces Hyperphantasia, a novel benchmark designed to evaluate the mental visualization capabilities of MLLMs, addressing a gap left by benchmarks focused on passive perception. The benchmark comprises four procedurally generated puzzle types: two requiring interpolation and two requiring extrapolation with increasing difficulty. Through comprehensive evaluation, the authors demonstrate that state-of-the-art MLLMs perform poorly, especially on harder tasks, revealing a significant gap compared to human ability and a tendency to rely on simplistic heuristics. Furthermore, the study uncovers a deeper failure in robust visual reasoning, as models often fail even when visual cues are explicitly provided, and shows that reinforcement learning on challenging examples can help foster more generalizable visualization skills.

**Dataset Code Accessibility:**

Yes

**Ethical Considerations:**

No, there are no or only very minor ethics concerns

**Final Justification:**

I appreciate the authors’ thorough rebuttal and constructive engagement during the discussion phase. My concerns are resolved so I keep my rating unchanged.

**Limitations Weaknesses:**

1. The paper claims a substantial gap between MLLMs and humans but it lacks a formal human performance baseline in its evaluations to quantify this gap.

2. The best-performing models are proprietary, which prevents a deeper analysis of why they succeed or fail, limiting the study to reporting outputs rather than explaining the underlying reasoning processes.

3. The benchmark consists exclusively of synthetic, 2D, abstract geometric puzzles, which may not generalize to the complex, 3D, and physics-based mental visualization required for real-world scenarios.

**Strengths Contributions:**

1. The paper presents Hyperphantasia, a new benchmark designed to evaluate the "mental visualization" capabilities of MLLMs. It moves beyond passive perception tests by using four procedurally generated puzzle types, each with three difficulty levels.

2. A comprehensive evaluation of state-of-the-art MLLMs on Hyperphantasia demonstrates a substantial performance gap compared to human abilities. Models struggle significantly as task difficulty increases, indicating that robust mental simulation and spatial reasoning remain a major challenge.

3.  The study finds that poor performance is not just due to a lack of mental visualization. In the experiment, models failed a puzzle even after the required lines were explicitly drawn, revealing a more fundamental inability to interpret slightly out-of-distribution visual patterns and highlighting the need for training on more challenging data.

---

> ### Author Rebuttal · Authors · 2025-07-31
>
> We thank the reviewer for their effort in reviewing our work. We are glad that the reviewer thinks our benchmark “moves beyond passive perception tests” and that our experiments are “comprehensive”. Please find below our response to the limitations and weaknesses.
> 1. Thank you for this valuable suggestion. We have obtained the required IRB approval and conducted a formal human evaluation (with at least 3 persons solving each puzzle). We will update the manuscript to include these results. We excluded the Seven Segments puzzles from this experiment as each digit is clearly recognizable based on its connections, making the task trivial for humans. Below, we present the results of our human evaluation alongside the accuracy of the best-performing model for each puzzle and difficulty level (CD: Connect the Dots, LT: Linear Trajectory, PT: Parabolic Trajectory):
>
>     |                   | CD Easy | CD Medium | CD Hard | LT Easy | LT Medium | LT Hard | PT Easy | PT Medium | PT Hard |
>     |-------------------|:----:|:------:|:----:|:----:|:------:|:----:|:----:|:------:|:----:|
>     |                   |      |        |      |      |        |      |      |        |      |
>     | Human             | 98.86 | 94.00 | 95.20 | 100 | 91.33 | 89.33 | 100 | 54.40 | 52.33 |
>     | The best model    | 96    | 80    | 67    | 60  | 29    | 30    | 40  | 34    | 33    |
>
>
>     These results highlight the gap between human and model performance. Even in the harder ball-trajectory problems, which are the most difficult problems of our benchmark, humans achieve around 50% accuracy by narrowing down the correct answer to two plausible options. In contrast, most models perform at or near random guessing, underscoring their lack of genuine mental visualization.
> 2. Thank you for bringing this up. We agree that the proprietary nature of the best-performing models limits full interpretability, which is a broader challenge shared by most benchmarks today, as these models continue to outperform open-source alternatives. Nevertheless, in our study, we partially analyze them through their reasoning traces. For instance, we observe hallucinated justifications in Claude's outputs, which reveal specific failure modes. Additionally, the correct reasoning patterns exhibited by these models can serve as valuable supervision signals for training and improving open-source models, a direction we believe is promising for future work.
> 3. Thank you for raising this point. In designing Hyperphantasia, our goal was to identify and isolate the foundational components needed for mental visualization. Our proposed puzzles capture the core basic skills that underlie the more complex real-world problems. For example, the extrapolation tasks reflect essential geometric reasoning capabilities necessary for understanding physical dynamics in real-world environments.
>
>     We believe that strengthening these basic capabilities is a necessary stepping stone toward models with robust mental visualization. That said, we view Hyperphantasia as a framework that can be naturally extended to richer, more diverse settings by adding new dynamics, more complex physical interactions, and multi-step prediction tasks. In future iterations, we plan to scale the benchmark along these axes, while maintaining the diagnostic clarity that comes from using controlled, interpretable setups.

---

> > ### Comment · Reviewer_cEcQ · 2025-08-05
> >
> > Thanks the author's response to my concerns. I do not have further questions and keep my rating unchanged.

---

> > > ### Author Response · Authors · 2025-08-07
> > >
> > > We appreciate the reviewer’s suggestions and valuable comments. We will use this feedback to improve the final version of the manuscript.

---

### Official Review · Reviewer_YzH2 · 2025-06-22

**Rating:** 4
**Confidence:** 3

**Summary:**

This paper introduces a novel dataset, Hyperphantasia, which is a benchmark for evaluating vision-language models on tasks of mental visualization. In particular, Hyperphantasia consists of four distinct puzzle types, including seven segment, connect the dots, triangle trajectory, and ball trajectory, spanning across three difficulty levels. There are 1200 samples. The authors also comprehensively evaluate existing SToA MLLMs on this benchmarks, using RL policy to fine-tune model on proposed dataset. The results showed the shortfall of existing MLLMs on mental visualization, compared to human.

**Dataset Code Accessibility:**

Yes

**Ethical Considerations:**

No, there are no or only very minor ethics concerns

**Final Justification:**

After reading the authors' response, I keep my initial rating.

**Limitations Weaknesses:**

## Dataset size may be limited.
* Current dataset is more likely a evaluation dataset. It is too small to be a training or finetuning dataset.

## Fairness of model comparison
* For different MLLMs, we need to use different strategies and prompts, which may also affect models' performance. How could we ensure the fairness of the model inference?

**Strengths Contributions:**

## Well-written
* This paper is well-written and easy to follow. Proposed datasets and tasks are clearly explained. The analysis of experiment results is extensive and thorough.

## Dataset is well-motivated and well-designed
* The construction of this dataset is well-motivated. Existing MLLM evaluation benchmarks typically focus on VQA, captioning, or perception task. Current dataset differs from them by providing partial visual observations to probe the models' ability on visual reasoning.
* The tasks in the datasets are also well-designed, across interpolation/extrapolation tasks, three level of difficulties, and various types of puzzles. These tasks not only make sense in its own way, but also easy to interpret.

## Comprehensive evaluation and analysis of existing MLLMs
* The authors evaluated a range of state-of-the-art MLLMs, including 4o, Gemini 2.5, Claude, Qwen, Deepseek-VL2, and so on. The evaluation results are interesting, revealing that most MLLMs fall short in visual reasoning and mental visualization. This finding further strengthes the significance of the proposed benchmarks.

## A simple baseline for improving mental visualziation
* Authors also pay effort to build a simple solution to improve the current models' performance on proposed task. This is a nice touch.

## Dataset and code are accessible

---

> ### Author Rebuttal · Authors · 2025-07-31
>
> We thank the reviewer for their valuable feedback on our manuscript. We are glad to hear that the reviewer finds the paper to be “well-written” and “well-designed”. Please find below our response.
> 1. Thank you for raising this concern. We agree that the 1200 samples in our benchmark are too small for large-scale training or fine-tuning. We note that these samples are only for evaluation, but we agree that having a large dataset for training and fine-tuning is important. To overcome this issue, we have included our puzzle generation scripts, which are fully automatic, to allow scaling the dataset for training purposes. This is exactly what we used for our RL experiments, where we generated around 10K samples for them. Overall, we believe one of the benefits of Hyperphantasia is that it can easily be scaled to generate enough training data, especially in higher difficulties, where training is more impactful. Finally, to address the reviewer’s concern, we will publicly release these large training datasets with around 10K samples for the community.
> 2. This is an insightful concern. While most current SOTA models are capable of following detailed instructions, ensuring that prompts are equally suitable across different models is crucial for a fair evaluation. To address this concern, we designed and tested four alternative prompts on the *easy Linear Trajectory puzzles* to assess the sensitivity of performance to prompt formulation. The prompts include: 1) A short and concise prompt, 2) A prompt that explicitly instructs the model for extra care and attention, 3) a prompt that only requests a single letter answer without CoT reasoning, and 4) a prompt where the models is introduced as an AI expert in mental visualization before being asked the question. Here are the results of these experiments:
>
>     | Model           | Original | Short | Careful | Single letter | AI expert |
>     |:---------------:|:--------:|:-----:|:-------:|:-------------:|:---------:|
>     | Qwen VL 2.5 7B  |    27    |  32   |   26    |      23       |     25    |
>     | Qwen VL 2.5 32B |    40    |  38   |   35    |      35       |     27    |
>     | Llama 3.2 11B   |    36    |  43   |   39    |      27       |     41    |
>     | Llama 3.2 90B   |    30    |  28   |   32    |      21       |     27    |
>     | LLaVA OV 7B     |    26    |  40   |   32    |      41       |     24    |
>     | LLaVA OV 72B    |    42    |  43   |   26    |      39       |     34    |
>
>     We note that during this experiment, we identified an error in the originally reported accuracy of LLaMA 3.2 in the manuscript. While we initially reported an accuracy of 23%, the correct value is 36%. This discrepancy was limited to this entry, and we have corrected it accordingly.
>
>     These results highlight that alternative prompts have a small effect on the performance and that our default prompt is not overly biased toward or against a particular model.

---

> > ### Comment · Reviewer_YzH2 · 2025-08-07
> >
> > The authors cleared my concerns in the rebuttal. I keep my original rating - boderline acceptance.

---

> > > ### Author Response · Authors · 2025-08-08
> > >
> > > We appreciate the reviewer’s effort in reviewing our paper and their thoughtful comments. We will do our best to incorporate all the feedback and further strengthen our work.

---

### Official Review · Reviewer_7r55 · 2025-07-03

**Rating:** 5
**Confidence:** 3

**Summary:**

The manuscript introduces Hyperphantasia, a novel benchmark designed to evaluate the mental visualization capabilities of multimodal large language models (MLLMs). The benchmark probes critical cognitive abilities such as interpolation, extrapolation, and visual reasoning through four synthetic puzzle types at varying difficulty levels. This work addresses a significant gap in current benchmarks, which focus more on passive visual perception rather than active visual simulation. While the paper makes a substantial contribution to the field, minor revisions are required to enhance clarity, reproducibility, and completeness.

**Dataset Code Accessibility:**

Yes

**Ethical Considerations:**

No, there are no or only very minor ethics concerns

**Final Justification:**

All my concerns have been addressed.

**Limitations Weaknesses:**

1. The RL experiments are promising but lack sufficient detail: you should specify the reward function used during RL training and how it incentivizes mental visualization, clarify the reasoning behind the chosen datasets for RL training (e.g., Easy vs. Medium puzzles) and their role in promoting generalization.
2. The paper highlights hallucination and heuristic-based reasoning as common failure modes but does not propose potential mitigation strategies. Thus you can discuss how visual grounding mechanisms or structured prompts could improve model performance.

**Strengths Contributions:**

1. The authors benchmark state-of-the-art MLLMs (e.g., GPT-4o, Gemini, Claude, Qwen) and reveal substantial performance gaps between humans and models, highlighting the challenges of mental visualization.
2. The benchmark includes four puzzle types (e.g., Seven Segments, Connect the Dots, Linear and Parabolic Trajectories) presented at three difficulty levels, offering a systematic evaluation of model capabilities.
3. This paper explores reinforcement learning to improve model performance on extrapolation tasks, demonstrating that carefully structured training can enhance generalization across difficulty levels.
4. The dataset and evaluation code are publicly available, ensuring transparency and encouraging adoption by the research community.

---

> ### Author Rebuttal · Authors · 2025-07-31
>
> Thank you for reviewing our paper and for the thoughtful comments. We are glad that the reviewer thinks that our benchmark offers a “systematic evaluation of model capabilities” and that it addresses “a significant gap in current benchmarks, which focus more on passive visual perception rather than active visual simulation”. Please find below our response to the reviewer's concerns.
> 1. Thank you for pointing this out. In our RL experiments, we use the standard GRPO reward formulation used in the literature. This reward has two components: formatting reward and accuracy reward. The formatting reward is 1 if the model’s response follows the \<think\>THINKING\</think\>\\boxed{ANSWER} format and 0 otherwise. Similarly, the accuracy score is 1 if the answer is correct and 0 otherwise. We will add these details to the appendix to ensure better clarity.
>
>     Regarding dataset selection, prior works such as [1] and [2] observed that the difficulty and quality of training samples can drastically improve the generalizability in the reasoning tasks. Our RL setup was designed to test this hypothesis directly. As we report in the paper, training on Medium-difficulty puzzles leads to stronger generalization than training on Easy puzzles, which often results in overfitting to superficial heuristics. We will clarify this motivation and highlight the connection to these prior works in the revision.
> 2. Thank you for this valuable suggestion. To explore possible mitigation strategies, we conducted new experiments testing the effects of multi-shot prompting and several alternative prompt formulations, including: 1) a short and concise prompt, 2) a prompt that explicitly instructs the model to take extra care, 3) a prompt that requests only a single-letter answer without CoT reasoning, and 4) a prompt that frames the model as an AI expert in mental visualization before presenting the question. Here are the results of these experiments:
>
>     Multi-shot prompting on the easy Linear Trajectory puzzles:
>     | Model                    | Zero-shot acc.     | 3-shot acc.         |
>     |-------------------------|:------------------:|:-------------------:|
>     | o4-mini                 |        43          |         40          |
>     | Qwen VL 2.5 7B          |        27          |         30          |
>     | Qwen VL 2.5 32B         |        40          |         31          |
>     | LLaVA OneVision 7B      |        22          |         29          |
>     | LLaVA OneVision 7B      |        42          | Could not answer    |
>     | Llama 3.2 11B           |        23          |         23          |
>     | Llama 3.2 90B           |        30          |         31          |
>
>    In the case of the larger LLaVA-OneVision model, it was unable to produce a meaningful answer when multiple images were included in the prompt. We speculate that this may be due to the lack of multi-image inputs in its training data. Interestingly, this issue does not arise with the smaller version of the model, which was able to provide valid answers.
>
>     The results of testing alternative prompts on the easy Linear Trajectory puzzles:
>
>     | Model           | Original | Short | Careful | Single letter | AI expert |
>     |:---------------:|:--------:|:-----:|:-------:|:-------------:|:---------:|
>     | Qwen VL 2.5 7B  |    27    |  32   |   26    |      23       |     25    |
>     | Qwen VL 2.5 32B |    40    |  38   |   35    |      35       |     27    |
>     | Llama 3.2 11B   |    36    |  43   |   39    |      27       |     41    |
>     | Llama 3.2 90B   |    30    |  28   |   32    |      21       |     27    |
>     | LLaVA OV 7B     |    26    |  40   |   32    |      41       |     24    |
>     | LLaVA OV 72B    |    42    |  43   |   26    |      39       |     34    |
>
>     We note that during this experiment, we identified an error in the originally reported accuracy of LLaMA 3.2 in the manuscript. While we initially reported an accuracy of 23%, the correct value is 36%. This discrepancy was limited to this entry, and we have corrected it accordingly.
>
>     The results showed minimal to no improvements in model performance and mitigating hallucination, implying that prompting techniques alone cannot resolve the underlying problem. Regarding visual grounding, we would appreciate further clarification on what the reviewer means. We are open to exploring this direction further and incorporating additional experiments in the revised manuscript.
>
> **References**
>
> [1] Xia, Jiaer, et al. *Visionary-R1: Mitigating Shortcuts in Visual Reasoning with Reinforcement Learning*. arXiv preprint arXiv:2505.14677, 2025.
>
> [2] Muennighoff, Niklas, et al. *s1: Simple Test-Time Scaling*. arXiv preprint arXiv:2501.19393, 2025.

---

> > ### Comment · Reviewer_7r55 · 2025-08-05
> > **Thank you for your rebuttal**
> >
> > All my concerns have been addressed, I will raise my scores!

---

> > > ### Author Response · Authors · 2025-08-07
> > >
> > > We thank the reviewer for their thoughtful feedback and support. We are pleased that all concerns have been addressed and will incorporate the suggestions to further strengthen the manuscript.

---

### Official Review · Reviewer_WX9J · 2025-07-07

**Rating:** 4
**Confidence:** 3

**Summary:**

This paper presents Hyperphantasia, an innovative benchmark designed to systematically evaluate mental visualization capabilities in multimodal large language models (MLLMs). Mental visualization - the cognitive ability to internally construct and manipulate visual representations - represents a fundamental aspect of human intelligence that enables complex spatial reasoning, dynamic trajectory prediction, and abstract visual problem solving. In contrast to existing benchmarks that focus primarily on passive visual perception tasks, Hyperphantasia introduces a carefully constructed suite of four distinct puzzle types, each featuring three progressive difficulty levels (totaling 1,200 samples). These tasks are strategically divided into interpolation challenges (e.g., completing partial geometric patterns) and extrapolation problems (e.g., predicting physical motion trajectories), requiring models to demonstrate genuine visual imagination beyond simple pattern recognition.

**Dataset Code Accessibility:**

Yes

**Dataset Code Comments:**

The url of codes are given on the first page.

**Ethical Comments:**

No, there are no or only very minor ethics concerns

**Ethical Considerations:**

No, there are no or only very minor ethics concerns

**Limitations Weaknesses:**

(1) The background section on Vision-Language Models (VLMs) suffers from inadequate coverage, providing only a limited survey of existing architectures. This incomplete literature review creates unnecessary difficulties for readers attempting to grasp the field's current landscape.

(2) The benchmark discussion should be substantially expanded to provide more forward-looking insights for future research directions. A well-designed benchmark should not only evaluate but also rank existing approaches, which requires more thoughtful dataset design considerations.

**Strengths Contributions:**

(1) The paper presents well-justified motivations and features a rationally designed dataset taxonomy with comprehensive coverage of relevant scenarios.

(2) The experimental evaluation is thorough, encompassing a diverse range of state-of-the-art vision models, thereby providing holistic insights into the research problem.

---

> ### Author Rebuttal · Authors · 2025-07-31
>
> We thank the reviewer for their valuable feedback on our manuscript. We are glad to hear that the reviewer finds the paper to have “well-justified motivations” and that our experimental evaluation is “thorough”. Please find below our response.
> 1. Thank you for this feedback. Based on the reviewer’s suggestion, we will update the background section of the manuscript to provide a better coverage of VLM architectures and their inner workings.
> 2. Thank you for raising this important point. We agree that benchmarks should not only highlight the limitations of the models but also propose potential solutions to address these limitations. In our manuscripts, we discussed Reinforcement Learning (RL) as one such solution, and our RL experiments aim to investigate its effectiveness.
>
>     Furthermore, based on the suggestions of other reviewers, we conducted additional experiments with multi-shot prompting (reviewer 9rXL) and alternative prompts (reviewer YzH2) including: 1) A short and concise prompt, 2) A prompt that explicitly instructs the model for extra care and attention, 3) a prompt that only requests a single letter answer without CoT reasoning, and 4) a prompt where the models is introduced as an AI expert in mental visualization before being asked the question. Here are the results of these experiments:
>
>     Multi-shot prompting on the easy Linear Trajectory puzzles:
>     | Model                    | Zero-shot acc.     | 3-shot acc.         |
>     |-------------------------|:------------------:|:-------------------:|
>     | o4-mini                 |        43          |         40          |
>     | Qwen VL 2.5 7B          |        27          |         30          |
>     | Qwen VL 2.5 32B         |        40          |         31          |
>     | LLaVA OneVision 7B      |        22          |         29          |
>     | LLaVA OneVision 7B      |        42          | Could not answer    |
>     | Llama 3.2 11B           |        23          |         23          |
>     | Llama 3.2 90B           |        30          |         31          |
>
>     In the case of the larger LLaVA-OneVision model, it was unable to produce a meaningful answer when multiple images were included in the prompt. We speculate that this may be due to the lack of multi-image inputs in its training data. Interestingly, this issue does not arise with the smaller version of the model, which was able to provide valid answers.
>
>     The results of testing alternative prompts on the easy Linear Trajectory puzzles:
>
>     | Model           | Original | Short | Careful | Single letter | AI expert |
>     |:---------------:|:--------:|:-----:|:-------:|:-------------:|:---------:|
>     | Qwen VL 2.5 7B  |    27    |  32   |   26    |      23       |     25    |
>     | Qwen VL 2.5 32B |    40    |  38   |   35    |      35       |     27    |
>     | Llama 3.2 11B   |    36    |  43   |   39    |      27       |     41    |
>     | Llama 3.2 90B   |    30    |  28   |   32    |      21       |     27    |
>     | LLaVA OV 7B     |    26    |  40   |   32    |      41       |     24    |
>     | LLaVA OV 72B    |    42    |  43   |   26    |      39       |     34    |
>
>     We note that during this experiment, we identified an error in the originally reported accuracy of LLaMA 3.2 in the manuscript. While we initially reported an accuracy of 23%, the correct value is 36%. This discrepancy was limited to this entry, and we have corrected it accordingly.
>
>     These experiments reveal that prompting techniques, while slightly helpful in some cases, have little to no effect on addressing the fundamental reasoning failures observed in our benchmark. This suggests that more substantial architectural or training-level changes are required.
>
>     Beyond RL and prompting techniques, we believe that a promising direction is to equip models with visual thinking capabilities. Current models reason entirely in the language domain, but Hyperphantasia puzzles and many real-world scenarios, the thinking cannot be explained with language, and they require visual thinking. While recent "omni" models offer some potential in this area, they are still significantly behind state-of-the-art language models in overall performance.
>
>     Importantly, we do not believe that omni models are the only path forward. Visual thinking tokens do not need to correspond to meaningful or interpretable images. Instead, they could function as internal visual representations. However, designing and training such models requires careful consideration, especially in how to cue or supervise this form of internal visualization. Some early efforts in this direction, such as [1], have explored visual reasoning in constrained setups, but these remain narrow in scope. We see Hyperphantasia as an ideal testbed to encourage progress in this direction.
>
>     We will incorporate these insights and future directions, as well as the results of the new prompting experiments, in the updated manuscript.
>
>     **References**
>
>     [1] Yang, Zeyuan, et al. "Machine Mental Imagery: Empower Multimodal Reasoning with Latent Visual Tokens." arXiv preprint arXiv:2506.17218 (2025).

---

> > ### Author Response · Authors · 2025-08-08
> >
> > We are grateful to the reviewer for their time, effort, and thoughtful suggestions. Based on the feedback, we conducted new experiments on the effects of prompting and expanded our discussion of future directions. We hope that our rebuttal answered all of the reviewer’s questions and concerns with respect to our paper. As the discussion period is closing soon, we kindly ask the reviewer to let us know if there are any further questions about our paper.

---

### Official Review · Reviewer_9rXL · 2025-07-21

**Rating:** 4
**Confidence:** 4

**Summary:**

Hyperphantasia tests the ability of VLMs to perform mental visualization, i.e., to internally construct or extrapolate visual structure without complete/ explicit pixel evidence. The benchmark comprises two categories (interpolation & extrapolation), each of which contains two tasks (for interpolation: digit inference from numeric sequence connections & 2) object inference from point-based contours; for extrapolation: inferring a linear trajectory & inferring a parabolic trajectory). Each of the four tasks contains 100 examples for each of the three difficulty levels (easy, medium, hard), totaling 1200 examples. The authors evaluate four SOTA proprietary VLMs, and a range of open-source models (7 B - 90 B params). The authors generally observe low accuracy for the extrapolation tasks, and relatively higher accuracy in the second interpolation task, whereas the first interpolation task is found to be solved with particularly high accuracy for one model and 0-accuracy for a majority of other models. The authors also explore using RL to train a 7B model on extrapolation task, and report improved performance when training on a mix of non-easy tasks.

**Additional Feedback:**

- it would be useful to include a breakdown/ taxonomy of errors that the models make, and inspect reasoning traces of models besides just  evaluating end-result accuracy
- (for the first interpolation task) it would be interesting to include ablations with text-only models to provide additional insights into how much can the performance derive from interpolative reasoning based on the text sequence only

**Dataset Code Accessibility:**

Yes

**Ethical Considerations:**

No, there are no or only very minor ethics concerns

**Final Justification:**

The authors have addressed my points -- I raise the score to borderline accept.

**Limitations Weaknesses:**

- it is not immediately clear how this dataset stands out compared to existing spatial reasoning and visualization datasets (e.g., Stogiannidis et al., 2025; Lee et al. 2025; etc).
- the extrapolation tasks are limited to only simple 2D trajectory inference; it is unclear to what degree can the dataset be further scaled and diversified
- the authors only include zero-shot evaluation, whereas the tasks at hand appear to require chain-of-thought-type reasoning; it is unclear to what degree could model performance improve with CoT/ in-context examples

**Strengths Contributions:**

- the paper is clearly written, easy to understand, and well motivated
- the benchmark stands out among the standard VQA/ visual reasoning benchmarks that commonly involve directly using the information accessible in the input image
- the authors evaluate an extensive range of both  models and expose failure modes of even the strongest proprietary models

---

> ### Author Rebuttal · Authors · 2025-07-31
>
> We thank the reviewer for their invaluable insights. We are glad that the reviewer finds that our work is “well motivated” and that our benchmark “stands out among the standard VQA/visual reasoning”. Please find below our response to the reviewer's concerns.
> 1. This is a great question. Hyperphantasia is designed to isolate and evaluate core mental visualization abilities without conflating them with other cognitive skills.  This focus enables us to find the exact failure modes of mental visualization in VLMs. In contrast, Stogiannidis et al. (2025) and Lee et al. (2025) incorporate additional layers of complexity, such as complex scene analysis or text recognition combined with an understanding of informal or slang language, making it difficult to attribute model failures specifically to weaknesses in mental visualization. Our benchmark consists of tightly controlled, abstract puzzles that target primitive skills, allowing for a more interpretable diagnosis of model limitations.
>
>     We would also like to note that both of the aforementioned benchmarks were released publicly after the NeurIPS submission deadline. While we could not include comparisons in the initial submission, we are happy to add a discussion of these works and their comparison to our benchmark in the final version of the manuscript.
> 2. Thank you for raising this point. As mentioned in the previous part, we intentionally designed the puzzles to focus on simple 2D tasks as a foundational skill. Our goal was to isolate and evaluate the core capabilities without confounding factors such as 3D reasoning, object dynamics, or complex physics. That said, we view Hyperphantasia as a framework that can be naturally extended to richer, more diverse settings by adding new dynamics, more complex physical interactions, and multi-step prediction tasks. In future iterations, we plan to scale the benchmark along these axes, while maintaining the diagnostic clarity that comes from using controlled, interpretable setups.
> 3. Thank you for this suggestion. Our current prompting format already encourages models to engage in CoT reasoning, and we observe that models indeed generate multi-step justifications when solving the puzzles. However, we agree that exploring multi-shot prompting is a valuable direction to better understand the effect of in-context learning on mental visualization tasks. To address this, we add an ablation study with multi-shot prompts to Hyperphantasia. We have already tested a number of models with 3-shot prompting on the easy samples of the linear trajectory puzzle using the prompt below:
>     ```
>     Each image shows a triangle composed of two blue edges and one red edge. In each image A portion of the red edge has been removed. The red line originates from the left red vertex and should connect to a point along the blue edge on the right side of the triangle. There are four possible points marked on this blue edge: A, B, C, and D.
>     Based on the direction and alignment of the red edge, we want to extrapolate the rest of the red line and determine which point (A, B, C, or D) is the vertex of the triangle corresponding to the missing part of the red edge.
>     The answer for the first three images is provided after them.
>     You should provide the answer for the fourth image only. The answer is one of the following: A, B, C, or D.
>     ```
>     Then we follow the prompt with 3 images and their corresponding answers that are randomly selected from the rest of the dataset (same puzzle and difficulty). At the end we show the image corresponding to the current sample followed by below instructions:
>     ```
>     You can explain your reasoning but you should specify your final answer by putting <ANSWER> before and after it like <ANSWER>FINAL_ANSWER<ANSWER> where FINAL_ANSWER is your answer. For example, if your answer is A, you should say <ANSWER>A<ANSWER>.
>     ```
>     Here are the results of the tested models:
>     | Model                    | Zero-shot acc.     | 3-shot acc.         |
>     |-------------------------|:------------------:|:-------------------:|
>     | o4-mini                 |        43          |         40          |
>     | Qwen VL 2.5 7B          |        27          |         30          |
>     | Qwen VL 2.5 32B         |        40          |         31          |
>     | LLaVA OneVision 7B      |        22          |         29          |
>     | LLaVA OneVision 7B      |        42          | Could not answer    |
>     | Llama 3.2 11B           |        23          |         23          |
>     | Llama 3.2 90B           |        30          |         31          |
>
>     In the case of the larger LLaVA-OneVision model, it was unable to produce a meaningful answer when multiple images were included in the prompt. We speculate that this may be due to the lack of multi-image inputs in its training data. Interestingly, this issue does not arise with the smaller version of the model, which was able to provide valid answers.
>
>     These results highlight the fact that although multi-shot prompting can slightly improve some models’ performance, it cannot address the underlying issue, implying that the models lack the mental visualization capabilities needed for these puzzles.
>
> **Responses to the feedback**
> 1. This is a great suggestion. We agree that analyzing the reasoning traces can provide deeper insights into the models’ behaviors. In fact, we already observe several interesting patterns in the responses. For example, in the Seven Segments tasks, several models answer with a fixed number (such as 012 in the case of Qwen VL 2.5 7B), or in the Connect the Dots puzzles, we find cases of hallucinating non-existing objects to justify an incorrect answer. These behaviours are briefly discussed in Section 4.2, but based on the reviewer’s feedback, we will expand this analysis and add a taxonomy of common error types across tasks in the final version of the manuscript.
> 2. We thank the reviewer for suggesting this interesting experiment. Based on this feedback, we designed a new prompt for the Seven Segment task in which we describe the grid in text, and we do not input an image anymore. The prompt below is the text-only prompt for this task:
>     ```
>     Solve the following puzzle:
>     Imagine a 3 by 6 grid of dots arranged as follows (rows: top to bottom, columns: left to right):
>     Column 1:  [0] [1] [2]
>     Column 2:  [3] [4] [5]
>     Column 3:  [6] [7] [8]
>     Column 4:  [9] [10] [11]
>     Column 5:  [12] [13] [14]
>     Column 6:  [15] [16] [17]
>     Now, imagine the following pairs of dots are connected with straight lines:
>     0 to 1, 0 to 3, 1 to 2, 1 to 4, 2 to 5, 3 to 4, 4 to 5, 6 to 9, 7 to 8, 7 to 10, 8 to 11, 9 to 10, 15 to 16, 16 to 17
>     What 3-digit number is formed by these connections?
>     You can explain your reasoning but you should specify your final answer by putting <ANSWER> before and after it like <ANSWER>FINAL_ANSWER<ANSWER> where FINAL_ANSWER is your answer.
>     ```
>     We tested some Language Models and some of the Vision Language Models used in our original experiments with this new text-only prompt with the Easy Seven Segment puzzles, and the results are as follows:
>     | Model (L: language-only, VL: vision-language capable)     | Normal prompt acc. | Text-only prompt acc.     |
>     |-----------------------------------------------------------|:------------------:|:--------------------------:|
>     | GPT-4o (VL)                                               |         4          |             1              |
>     | O4-mini (VL)                                              |        83          |            88              |
>     | Qwen VL 2.5 7B (VL)                                       |         0          |             0              |
>     | Qwen VL 2.5 32B (VL)                                      |         0          |              1             |
>     | Qwen 2.5 32B Instruct (L)                                 |       N/A          |             0              |
>     | DeepSeek-R1 Distill Qwen 32B (L)                          |       N/A          |     0 (keeps thinking)      |
>     | Llama 3 8B Instruct (L)                                   |       N/A          |             0              |
>     | DeepSeek-R1 Distill Llama 8B (L)                          |       N/A          |             0              |
>
>     Based on these results, we can see that o4-mini, which was the best model on this task, can still solve them with high accuracy, with even a slight improvement. Even in the case of GPT-4o, despite its low accuracy, it has a lot of answers with only one incorrect digit, suggesting that it can partially solve the puzzles. However, the Language-only Models completely fail this task, indicating that they lack the required mental visualization capabilities. This outcome is expected, given that these models have not been trained on any visual data. An interesting case is Qwen VL 2.5 32B, which correctly solves one puzzle, where the answer is "118"; however, this may be due to the model's bias toward using the digits 1 and 8 in its responses.

---

> > ### Author Response · Authors · 2025-08-08
> >
> > We appreciate the reviewer’s comments on our paper, and we did our best to incorporate all the feedback and further strengthen our work. We added new experiments following the reviewer’s exciting ideas of multi-shot prompting and testing language-only models with the Seven Segments puzzles. We hope that the reviewer finds our rebuttal and the additional updates to the manuscript satisfactory. As the end of the rebuttal period is closing in, we would like to ask whether the reviewer has any lingering concerns or questions with respect to our manuscript.

---

> ### Comment · Area_Chair_DDXq · 2025-08-08
>
> Dear Reviewer 9rXL,
>
> Please help go through the rebuttal and participate in discussions (at least acknowledge you read the rebuttal) with authors. Thank you!
>
> Best regards, AC

---

### Note · Authors · 2025-08-14

We appreciate the time and effort of all reviewers and their insightful feedback. In response, we carefully addressed the raised concerns and conducted several new experiments and analyses to strengthen the paper. Below is a summary of the key changes and additions made during the rebuttal phase:
1. **Multi-shot prompting experiments** (reviewer 9rXL suggestion): We conducted multi-shot prompting experiments on a subset of puzzles. The results showed minimal improvements, suggesting that such prompting techniques are insufficient to improve mental visualization capabilities.
2. **Prompt sensitivity experiments** (reviewer YzH2 suggestion): We curated four alternative prompts to evaluate whether prompt design affects performance. The results demonstrated limited impact, reinforcing the need for deeper architectural or training improvements rather than prompt tuning.
3. **Human evaluation of Hyperphantasia** (reviewer cEcQ suggestion): We obtained IRB approval and conducted a formal human study, with at least three participants per puzzle. The results show a large gap between human performance and SOTA VLMs.
4. **Seven Segment puzzles in a fully language format** (reviewer 9rXL suggestion): We redesigned the Seven Segment puzzles as purely language-based tasks to test language-only models. The results show that the VLMs that were previously able to solve this task can still solve it, but the other models still fail. Moreover, the language-only models also fail as expected since they have not seen any visual data.
5. **Releasing a larger version of Hyperphantasia** for training and fine-tuning purposes (reviewer YzH2).
6. **Expanded Background and Future Directions** (reviewer WX9J): We will enrich the background section with additional details on VLM architectures and add a comprehensive discussion of future directions, including reinforcement learning and equipping models with visual thinking capabilities.

---

### Decision · Program_Chairs · 2025-09-18

**Decision:**

Accept (poster)

**Comment:**

This paper introduces Hyperphantasia, a benchmark designed to evaluate the mental visualization capabilities of multimodal large language models (MLLMs). The reviewers find the paper well-motivated, clearly written, and supported by comprehensive experiments on both proprietary and open-source models, which reveal substantial performance gaps and expose important failure cases.

During the rebuttal stage, the authors have added new experiments (multi-shot prompting, alternative prompts, text-only puzzles, and human evaluation), clarified methodological details, and expanded the discussion of limitations and future directions. While the tasks are still limited to synthetic 2D puzzles and interpretability remains a challenge, the benchmark provides a valuable controlled testbed and a foundation for future extensions to richer scenarios. Overall, the paper makes a timely and significant contribution to the evaluation of reasoning and visualization in MLLMs.